# Median Nerve Stimulation Facilitates the Identification of Somatotopy of the Subthalamic Nucleus in Parkinson’s Disease Patients under Inhalational Anesthesia

**DOI:** 10.3390/biomedicines10010074

**Published:** 2021-12-30

**Authors:** Yu-Chen Chen, Chang-Chih Kuo, Shin-Yuan Chen, Tsung-Ying Chen, Yan-Hong Pan, Po-Kai Wang, Sheng-Tzung Tsai

**Affiliations:** 1Department of Neurosurgery, Hualien Tzu Chi Hospital, Buddhist Tzu Chi Medical Foundation, Hualien 970, Taiwan; spring810569@gmail.com (Y.-C.C.); william.sychen@msa.hinet.net (S.-Y.C.); peterpan@gmail.com (Y.-H.P.); 2Department of Medical Informatics, Tzu Chi University, Hualien 970, Taiwan; 3Department of Physiology and Master Program in Medical Physiology, Tzu Chi University, Hualien 970, Taiwan; cckuo@mail.tcu.edu.tw; 4School of Medicine, Tzu Chi University, Hualien 970, Taiwan; chenyting@mail.tcu.edu.tw; 5Department of Anesthesiology, Hualien Tzu Chi Hospital, Buddhist Tzu Chi Medical Foundation, Hualien 970, Taiwan

**Keywords:** deep brain stimulation, general anesthesia, inhalational anesthesia, median nerve stimulation, neural signal analysis, Parkinson’s disease, subthalamic nucleus

## Abstract

Deep brain stimulation (DBS) improves Parkinson’s disease (PD) symptoms by suppressing neuropathological oscillations. These oscillations are also modulated by inhalational anesthetics used during DBS surgery in some patients, influencing electrode placement accuracy. We sought to evaluate a method that could avoid these effects. We recorded subthalamic nucleus (STN) neuronal firings in 11 PD patients undergoing DBS under inhalational anesthesia. Microelectrode recording (MER) during DBS was collected under median nerve stimulation (MNS) delivered at 5, 20, and 90 Hz frequencies and without MNS. We analyzed the spike firing rate and neuronal activity with power spectral density (PSD), and assessed correlations between the neuronal oscillation parameters and clinical motor outcomes. No patient experienced adverse effects during or after DBS surgery. PSD analysis revealed that peripheral 20 Hz MNS produced significant differences in the dorsal and ventral subthalamic nucleus (STN) between the beta band oscillation (16.9 ± 7.0% versus 13.5 ± 4.8%, respectively) and gamma band oscillation (56.0 ± 13.7% versus 66.3 ± 9.4%, respectively) (*p* < 0.05). Moreover, 20-Hz MNS entrained neural oscillation over the dorsal STN, which correlated positively with motor disabilities. MNS allowed localization of the sensorimotor STN and identified neural characteristics under inhalational anesthesia. This paradigm may help identify an alternative method to facilitate STN identification and DBS surgery under inhalational anesthesia.

## 1. Introduction

Parkinson’s disease (PD) is a neurodegenerative disease characterized by loss of dopaminergic neurons, which causes abnormalities in the downstream basal ganglia such as abnormal firing patterns (e.g., bursting activities) and irregular oscillatory activity. Subthalamic nucleus (STN) deep brain stimulation (DBS) is an effective treatment for PD, possibly exerting its effects via various plausible mechanisms [1,2]. Neurophysiological mapping with microelectrode recording (MER) during STN-DBS is an essential procedure for optimizing the placement of DBS electrodes [3]. The somatotopic organization of the STN has been revealed with MER, and these findings suggest that implanting the DBS electrode into the motor subterritory of the STN (i.e., dorsolateral STN) helps improve motor disability in PD. On the other hand, electrically stimulating the ventrolateral STN, including the limbic and associative neural circuits, may cause various neuropsychiatric effects.

DBS surgery is typically performed under local anesthesia while the patient remains awake in order to ensure accurate electrophysiological mapping. This approach allows physicians to conduct intraoperative microstimulation tests to assess motor responses with minimal adverse effects. However, if a PD patient is experiencing marked off-medication symptoms such as anxiety, painful dystonia, and respiratory distress, the patient may not be amenable to this lengthy surgical procedure [4]. In such cases, using general anesthesia during STN-DBS is an alternative approach. With this approach, MER is essential for monitoring patient symptomatology in order to ensure optimal target placement. Previous studies [5,6,7] have shown that anesthetic agents (e.g., propofol and dexmedetomidine) affect the background spontaneous firing and neuronal spike activity patterns of the basal ganglia. Moreover, we have previously [8] revealed that general anesthesia using the inhalational anesthetic sevoflurane decreases beta-frequency oscillations. One study [9] has indicated differences between desflurane and sevoflurane in terms of analgesia and hypnosis, although no significant difference was found in beta-band oscillation. Despite this finding, previously published studies and our own research [10,11,12,13,14] showed similar efficacy of DBS performed under local anesthesia and under general anesthesia in terms of clinical outcomes among patients with PD. We have previously demonstrated the feasibility of conducting MER during conditioned inhalational anesthesia with desflurane, which yielded good long-term outcomes [15].

Somatotopic mapping within the STN during MER may facilitate optimal placement of DBS electrodes. Nevertheless, this procedure is time-consuming, particularly under inhalational anesthesia, and the STN topography may not be delineated by MER signals due to partial suppression by anesthetics [7]. This may result in more stimulation-related adverse effects during the clinical follow-up. We considered that in addition to determining the correlation between the STN topography and the characteristic MER signal, enhancing MER neuronal signals by non-invasive median nerve stimulation (MNS) under conditioned inhalational anesthesia may help improve DBS outcomes. We hypothesized that various parameters from MNS could aid in the identification and differentiation of neuronal responses over the dorsal and ventral STN under inhalational anesthesia.

Thus, in this study, we aimed to identify stimulation parameters and strategies by using peripheral MNS that would make it possible to avoid the unnecessary adverse effects of inhalational anesthesia in patients undergoing DBS surgery for PD; Figure 1 displays a graphical abstract of the study.

## 2. Results

### 2.1. Clinical Outcomes after STN-DBS and Localization

The characteristics of the PD patients are shown in Table 1.

Intraoperative recordings from 26 MER units in the dorsal parts and 29 units in the ventral parts were analyzed for spontaneous STN activity. All patients had similar motor disabilities before undergoing bilateral STN-DBS. Their symptoms were significantly improved by DBS treatment (Table 2).

The effectiveness of STN-DBS did not differ among patients. Postoperative DBS substantially improved the clinical status of the patients. A follow-up examination at 14.0 ± 2.6 months after DBS surgery revealed that STN-DBS significantly improved the Unified Parkinson’s Disease Rating Scale (UPDRS) Part II scores (percentage of improvement from off-medication vs. on-medication, 55.7% ± 21.8%; *p* < 0.001; percentage of improvement from DBS off vs. DBS on, 57 ± 15.4; *p* = 0.002) and Part III scores (percentage of improvement from off-medication and on-medication, 45% ± 12.1%; *p* < 0.001; percentage of improvement from DBS off vs. DBS on, 43 ± 9.7; *p* < 0.001).

### 2.2. Analysis of Dorsolateral and Ventromedial STN Activity

The neuronal spike firing rate characteristics are listed in Table 3. The mean and standard derivation firing rate along the dorsal part of the STN were 54.9 ± 24.4 Hz, 57.6 ± 24.2 Hz, 57.8 ± 26.8 Hz, and 56.8 ± 24.3 Hz during no MNS, 5-Hz MNS (MNS-5), 20-Hz MNS (MNS-20), and 90-Hz MNS (MNS-90), respectively. Firing rates of 60 ± 30.3 Hz, 58.4 ± 35.3 Hz, 57.7 ± 34.1 Hz, and 52.8 ± 36.0 Hz were recorded during no-MNS, MNS-5, MNS-20, and MNS-90, respectively, within the ventral part of the STN. The firing rates were not significantly different between no-MNS and each of the MNS conditions in the dorsolateral and ventromedial regions.

We examined the oscillatory features of the MER signals using power spectrum density (PSD), which was divided into theta-, alpha-, beta-, and gamma-band oscillations (Figure 2).

When we compared the no-MNS condition with each of the MNS conditions for the whole STN, we found no significant differences. We then compared the dorsolateral and ventromedial STN activity under the different MNS conditions. In theta-band oscillations, a significant difference was found between no-MNS and MNS-5. In the alpha-band oscillations, a significant difference was found between the no-MNS, MNS-5, and MNS-20 conditions. The gamma-band oscillations differed significantly across all conditions. Moreover, we calculated the difference in percentage between the dorsal and ventral parts. The largest difference between beta- and gamma-band oscillations was observed for MNS-20 (beta: 4.2%; gamma: −10.7%), while the no-MNS condition showed a larger difference in theta- and alpha-band oscillations (theta: 2.8%; alpha: 3.1%) (Figure 3).

### 2.3. Correlation of PSD and Clinical Outcomes

Spearman’s test was used to evaluate correlations between individual power bands in the dorsal or ventral parts of the STN under inhalational anesthesia with different MNS parameters and motor disabilities (Table 4). We did not find a significant correlation between neuronal activity and motor clinical outcomes with the no-MNS condition. In contrast, Part II scores showed a significant positive correlation with beta-band oscillations in the dorsal STN under the MNS-5 (ρ = 0.71, *p* = 0.023), MNS-20 (ρ = 0.83, *p* = 0.003), and MNS-90 (ρ = 0.80, *p* = 0.005) conditions. Moreover, these scores showed a negative correlation with gamma-band oscillations under the MNS-20 condition. In the ventral STN, only gamma-band oscillations under the MNS-5 condition showed a negative correlation with Part II scores (ρ = −0.693, *p* = 0.026). The Part III scores were strongly correlated with beta-band oscillations under the MNS-5 and MNS-20 conditions, and no correlation in the ventral STN. Firing rates were not correlated with any neurological symptoms.

## 3. Discussion

In this study, we sought to identify new DBS stimulation parameters and strategies for PD patients by using noninvasive peripheral MNS to avoid unnecessary adverse effects of inhalational anesthesia in PD patients undergoing DBS surgery. We applied MNS at different frequencies (5 Hz, 20 Hz, and 90 Hz) and observed neuronal activity during DBS under inhalational anesthesia. The firing rate showed a slight increase in the PDS in the dorsal STN for all MNS frequencies. In contrast, a nonsignificant decrease in firing rate occurred in the ventral STN under all MNS conditions.

We analyzed the PSD to evaluate the effects of MNS at different frequencies on the oscillation frequency bands within the STN. The neuronal oscillations with no MNS under inhalational anesthesia and inhalational anesthetics were significantly different between the dorsal and ventral parts in terms of theta-, alpha-, and gamma-band oscillations, but not beta-band oscillations. MNS-20 produced beta-band oscillations with significantly higher power in the dorsal STN than in the ventral STN; this phenomenon did not occur under the MNS-5 and MNS-90 conditions. Beta-band oscillations (13–30 Hz) are the main feature distinguishing the dorsolateral STN and can be used to guide DBS electrode placement under local or inhalational anesthesia [16,17,18,19]. Previous studies have revealed that the pathological oscillations and the characteristics of neuronal oscillations in the STN of PD patients may change under general anesthesia [6,7,20]. Previous research further indicates that neuronal activity differs significantly in terms of beta-band oscillations between the dorsal and ventral parts of the STN under local anesthesia, with no significant differences in beta-band oscillations under inhalational anesthesia [8,9,21]. Only beta-band oscillations under the MNS-20 condition were the same as those under local anesthesia. Moreover, previous studies have demonstrated that gamma-band oscillations are higher within the ventromedial STN under DBS [19,22,23,24]. In this study, we demonstrated that both in the absence of MNS and for all frequencies of MNS there were significant differences in gamma-band oscillations between the dorsal and ventral STN, with the largest differences observed under the MNS-20 condition.

An earlier study [25] demonstrated that motor symptoms (e.g., limb rigidity and bradykinesia) in PD were correlated with beta-band oscillations in the STN under local anesthesia, while gamma-band oscillations were correlated with bradykinesia. In addition, the local field potential and PSD of beta-band oscillations was correlated with the degree of improvement in bradykinesia and rigidity, although not with tremors, after dopamine medication treatment [26,27,28]. These studies demonstrate that there is a correlation between neuronal oscillation and the severity of different motor symptoms under local anesthesia. In the UPDRS, Part II is used to evaluate motor experiences in daily living, and Part III is a motor examination that includes bradykinesia, limb rigidity, axial symptoms, tremor, etc. [29]. In our previous study [30], we demonstrated a lack of association between STN neuronal activity parameters (e.g., spectral density of the four band powers) and motor symptoms (e.g., rigidity, bradykinesia, axial symptoms, and tremor) under inhalational anesthesia. The phenomenon proved that anesthetics affected the neurophysiological features of PD. In our analysis of inhalational anesthesia without MNS, there was no correlation of Part II and Part III scores with the beta- and gamma-band oscillations throughout the STN. After MNS, we observed an association of Part II scores with beta-band oscillations. In addition, Part III scores were correlated with beta-band oscillations under MNS-5 and MNS-20 conditions. In contrast, no correlation between neuronal activity and motor symptoms existed in the ventral STN. Previous studies have shown that motor symptoms of PD are related to the dorsal STN [31,32]. In our study, neuronal activity in the ventral STN was not coherently induced by MNS. We did not find any association between neuronal activity characteristics and clinical motor symptoms under inhalational anesthesia. However, MNS facilitated not only the identification of neurophysiological signatures of the STN, but more clearly showed the correlation between motor symptoms and neuronal oscillation. In particular, MNS-20 may strengthen the correlation between motor symptoms and beta-band oscillation in the dorsal parts under inhalational anesthesia.

Our study had several limitations. First, we did not include a control group receiving MNS without inhalational anesthesia (i.e., with local anesthesia during DBS) to compare differences in neural firings. However, our included patients underwent MNS under different conditions, but with the same anesthesia depth. This could help to elucidate how inhalational anesthetics have different effects on neural oscillation under different peripheral stimulation parameters. Second, we used desflurane or sevoflurane to maintain general anesthesia during neural recording of STN. We did not perform sub-group analysis between the different anesthetics due to limited case numbers. Finally, as this was a small case series study, the findings may be confounded by interpersonal variability. Future research with more patients might provide direct evidence without heterogeneity, along with additional insight into how peripheral neural stimulation modulates intracranial recordings and oscillations under various inhalational anesthetics.

## 4. Methods

### 4.1. Patients

Eleven consecutive patients with PD who underwent bilateral STN-DBS from March 2018 were enrolled in this study. The study was conducted in accordance with the Declaration of Helsinki, and the protocol was approved by the Ethics Committee of Tzu Chi General Hospital, (Hualien, Taiwan; approval no. IRB105-17-A). All patients gave their informed consent for inclusion before they participated in the study.

All PD patients met the United Kingdom Parkinson’s Disease Brain Bank diagnostic criteria and demonstrated at least two main symptoms. Each patient underwent a levodopa test prior to surgery to confirm a positive levodopa response (>30% improvement in UPDRS Part II and Part III scores). Each patient underwent brain magnetic resonance imaging (MRI) preoperatively to rule out structural abnormalities. All patients underwent evaluation using the UPDRS in four different conditions: pre-operative on medication (Med on), pre-operative off-medication (Med off), post-operative off medication off DBS (Med off/DBS off) and post-operative on medication on DBS (Med on/DBS on). Details regarding the evaluation procedures (including UPDRS part II and part III) are described in our previous report [8,32,33]. The extent of medication improvement was evaluated with Med on and compared with Med off. The improvement after surgery was calculated with Med off DBS off and Med on DBS on status.

### 4.2. Imaging and Targeting

Images were obtained using a 1.5-T MRI scanner (General Electric Healthcare, Chicago, IL, USA). The standard settings included T1-weighted axial images of 0.75-mm slice thickness and T2-weighted axial images of 2-mm slice thickness. Each sequence was conducted with contiguous slices. The images were transferred to the Digital Imaging and Communications in Medicine database and the Stealth Neuronavigation workstation (Medtronic, Minneapolis, MN, USA). Image fusion software was used to fuse the two sets of MR images using 3D reconstruction. The tentative surgical target coordinates for placement of the tip of the permanent implantable electrode were set at the central and lowest border of the STN by direct visualization on MRI, as previously described [33]. A Leksell G-frame unit (Elekta Instrument, Inc., Atlanta, GA, USA) was used for the stereotactic procedure. The patient was placed in a straight supine position, and the head frame was secured in a Mayfield adaptor. The target coordinates were applied to the stereotactic frame and the working stage.

### 4.3. Anesthetic Procedure

All patients received inhalational anesthesia via endotracheal intubation. Anesthesia was initially induced by administering regular narcotic agents. In all patients, anesthesia was maintained using desflurane or sevoflurane inhalation during the surgical procedures. The depth of anesthesia was maintained at 0.5–1.0 minimal alveolar concentration, and each patient’s heart rate and blood pressure were monitored to ensure the absence of a cough reflex as well as changes in heart rate and blood pressure during the MER procedure.

### 4.4. Electrical Stimulation of the Median Nerve

A constant current stimulator (model DS7A, Digitimer Ltd., Letchworth Garden City, UK) was used to apply electrical stimulation to the contralateral median nerve during MER. A stimulation electrode was placed on the wrist (cathode: median nerve, 2-cm proximal to the wrist crease; anode: 2-cm distal to the cathode). The stimulation parameters included a pulse width of 0.2 ms, an intensity of 30 mA, and frequencies of 5 Hz (MNS-5), 20 Hz (MNS-20), and 90 Hz (MNS-90).

### 4.5. MER Procedure

The microelectrode was 10–40 μm in diameter and 200 mm in length and had a <50-μm tungsten tip and recording impedance between 0.5 MΩ and 1 MΩ. The microelectrode signal was recorded using an intraoperative MER system (LeadPoint; Medtronic, Fridley, MN, USA) where the signal was amplified (×10) and filtered (300–3 kHz). Recording started at 10 mm above the planned target coordinates. The microelectrode was advanced in steps of 200–500 µm, with pauses at sites of robust neuronal firing. Firing at each depth was recorded for the no-MNS, MNS-5, MNS-20, and MNS-90 conditions. The latency of discharge from each depth was recorded for 10 s.

### 4.6. Localization of Active Contact with Postoperative Computed Tomography

Consecutive axial computed tomography slices of the brain (thickness, 1.25 mm) were obtained after surgery to exclude intracranial complications and localize the postoperative electrode coordinates through image fusion with the preoperative MRI [33]. An acute stimulation test was conducted at 1 week post-surgery to select the optimal stimulation contact and parameters for chronic stimulation.

### 4.7. Data Processing and Analysis

MER traces were excluded based on the following conditions: (1) the duration of the recording trace was <10 s; and (2) the recording contained artifacts with amplitudes exceeding 300 µV or arm movements at the onset of the baseline “rest” condition. Each MER trace was classified as a dorsal or ventral trace based on the central point in STN-in and STN-out. To prove that the microelectrode recordings were under MNS, we extracted MER data about 4 to 8 s for analysis. In addition, we applied a notch filter to remove artifacts from the MNS which removed 5 Hz, 20 Hz, and 90 Hz signals under the MNS-5, MNS-20, and MNS-90 conditions, respectively.

### 4.8. Firing Rate and PSD Estimation

We detected a spike train by a threshold that was an amplitude >3.5 standard deviations every 1 s of recording and evaluated the firing rate over the entire session using 1-s bins. We analyzed the percentage change in firing rates at different MNS stimulation frequencies. Records were enveloped using the full-wave rectification method. We applied the PSD of the spike train with Thomson’s multi-taper method [34] to evaluate the neuronal oscillation characteristics. In addition, parameters of Thomson’s multi-taper method included the time half-bandwidth product of 3, 50% overlap between windows, and the number of samples with 3 s that produced 1/3 Hz spectral resolution. Each PSD was normalized by integrating the 3–100-Hz band (excluding the 48–52-Hz band) to obtain the relative power within the band. We then determined the normalized spectral power from the theta- (3–8 Hz), alpha- (8–13 Hz), beta- (13–30 Hz), and gamma- (30–100 Hz) band ranges in each trace. To analyze the topographical distribution of STN spike properties, the STN was divided into dorsal (0–50%) and ventral (50–100%) components.

### 4.9. Statistical Analysis

Statistical analyses were conducted using SPSS software 21 (IBM Corp., Armonk, NY, USA) and MATLAB 2019 (Mathworks, Inc., Natick, MA, USA). The Mann–Whitney *U* test was used to compare between no-MNS and the different frequencies of MNS, with *p* < 0.05 considered significant. Moreover, to evaluate the discrepancy between the dorsal and ventral parts in the four different frequency MNS conditions, Spearman correlations were used to estimate the association between STN neuronal activity parameters (i.e., spectral density of beta and gamma) and UPDRS scores (Part II and Part III) during off-medication. Statistical significance was set at *p* < 0.05.

## 5. Conclusions

Anesthetics affect the neurophysiological features of PD under inhalational anesthesia, which may cause inaccurate placement of an implant electrode during DBS surgery. However, we found that MNS at 20 Hz may increase beta-band oscillations in the dorsal STN and increase gamma-band oscillations in the ventral STN to facilitate appropriate placement of the stimulating electrode during DBS surgery. Moreover, we showed that MNS may strengthen neuronal activity in the dorsal STN, which is related to motor symptoms.

## Figures and Tables

**Figure 1 biomedicines-10-00074-f001:**
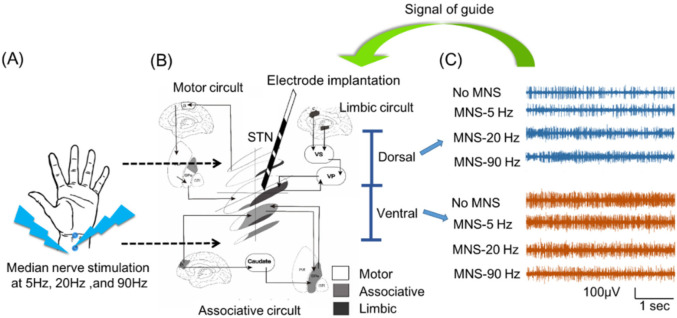
Graphical abstract of this study. (**A**) Patients with Parkinson’s disease (PD) underwent surgery for deep brain stimulation (DBS) with inhalation anesthesia; a stimulation electrode was placed on the wrist with a pulse width of 0.2 ms, an intensity of 30 mA, and frequencies of 5 Hz (MNS-5), 20 Hz (MNS-20), and 90 Hz (MNS-90). (**B**) Recording of every depth of STN for 10 s trace during stimulation with no MNS, MNS-5, MNS-20, and MNS-90. Each trace was classified as a dorsal or ventral trace on the central point for both STN-in and STN-out. (**C**) The neuronal oscillation characteristics under MNS with various frequencies were analyzed in order to potentially provide an appropriate signal of guide for electrode implantation during DBS surgery in PD patients under inhalation anesthesia.

**Figure 2 biomedicines-10-00074-f002:**
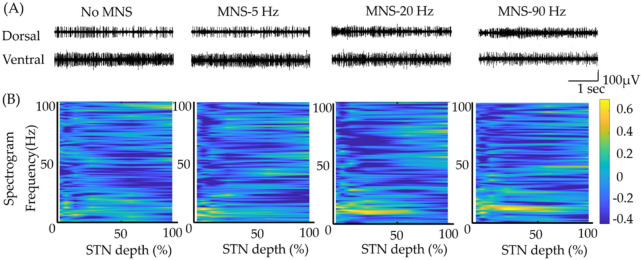
The analysis of power spectral density (PSD) in the dorsal and ventral subthalamic nucleus (STN) under no MNS and various frequencies of MNS. (**A**) Two representative microelectrodes recorded data from the dorsal and ventral STN under no MNS and various frequencies of MNS. (**B**) Topographical spectrogram changes revealed higher values in the beta band oscillation in dorsal STN and gamma band oscillation in ventral STN under MNS−20.

**Figure 3 biomedicines-10-00074-f003:**
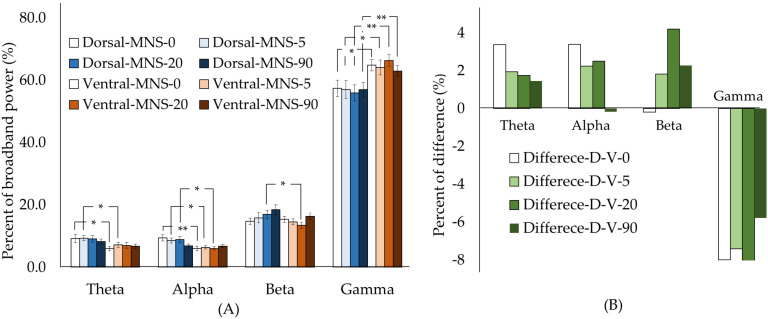
Comparison of each frequency band between dorsal and ventral STN under different frequencies of MNS and no MNS. (**A**) The moderately significant differences between dorsal and ventral STN were in the gamma oscillation band with MNS−5 and MNS−90, and in the alpha oscillation band with no MNS. (**B**) The percent differences in each frequency band between dorsal and ventral STN under different frequency MNS and no MNS show the largest difference in the theta and alpha oscillation bands, while MNS−20 shows the largest difference in the beta and gamma oscillation bands. Single asterisk (*) indicates a significant difference with *p* value < 0.05. Double asterisks (**) indicate a moderately significant difference between dorsal and ventral STN with *p* value < 0.01. MNS−0 indicates no MNS.

**Table 1 biomedicines-10-00074-t001:** Subject characteristics of Parkinson’s disease patients undergoing deep brain stimulation.

Age at onset (years)	48.0 ± 5.9
Age at surgery (years)	58.0 ± 6.6
Sex (male/female)	7/4
Height (cm)	162.1 ± 8.7
Weight (kg)	66.6 ± 15.6
BMI (kg m^−2^)	25.1 ± 4.6
Disease duration (years)	10.0 ± 2.9
Follow-up (months)	14.0 ± 2.6

Data are expressed as frequencies for categorical variables, mean ± standard deviation for age at onset, age at surgery, and disease duration, height, weight, body mass index, and follow-up.

**Table 2 biomedicines-10-00074-t002:** Pre-operative improvement from medication and post-operative medication; deep brain stimulation synergistic effectiveness.

	Med off	Med on	Improvement (%) ^1^	*p* Value	Med off/DBS off	Med on/ DBS on	Improvement (%) ^2^	*p* Value
Part II	21.4 ± 7.4	8.5 ± 2.8	55.7 ± 21.8	<0.001	21.0 ± 8.3	9.7 ± 6.5	61.9 ± 15.4	0.002
Part III	41.5 ± 8.0	22.5 ± 5.1	45.0 ± 12.1	<0.001	45.5 ±10.1	28.1 ± 8.3	43.9 ± 9.7	<0.001
Brady	17.6 ± 4.0	11.1 ± 3.4	35.9 ± 18.7	<0.001	20.7 ± 5.6	14.8 ± 4.6	34.2 ± 16.5	<0.001
Tremor	2.9 ± 2.1	0.3 ± 0.7	68.3 ± 43.4	0.001	2.2 ± 2.1	0.1 ± 0.3	76.7 ± 41.7	0.003
Rigidity	9.4 ± 2.2	3.9 ± 2.3	58.4 ± 20.4	<0.001	11.9 ± 2.4	5.8 ± 2.1	61.6 ± 11.5	<0.001
P&G	4.4 ± 0.8	2.7 ± 1.1	37.0 ± 25.8	<0.001	4.1 ± 0.7	2.5 ± 1.0	43.3 ± 11.0	<0.001
Axial	9.6 ± 2.1	5.9 ± 1.6	36.3 ± 21.2	<0.001	9.2 ± 2.8	6.4 ± 1.9	31.9 ± 13.2	0.009
H & Y	3.0 ± 0.7	2.8 ± 0.3			3.5 ± 0.7	3.0 ± 0.4		0.025

Data are expressed as mean ± standard deviation. Student’s *t*-test was used for statistical analysis of clinical outcomes under Med off, Med on, and DBS during Med off and during Med on. H&Y: Hoehn and Yah; P&G: Posture & Gait; Med off: medication off, Med on: medication on, DBS on: deep brain stimulation on; DBS off: deep brain stimulation off. The extent of improvement (%) ^1^ was first calculated from individual patient’s Med off scores–Med on scores/Med off scores × 100%; the value was the mean and standard deviation calculated from each patient’s improvement percentage in scores. Improvement (%) ^2^ was the mean of the percentage of DBS improvement, calculated from Med off and DBS off scores—Med on DBS on scores/Med off and DBS off scores × 100%.

**Table 3 biomedicines-10-00074-t003:** The firing rate under no MNS, MNS-5, MNS-20, and MNS-90; Student’s t-test was used to compare the differences between no MNS and different frequencies of MNS. In addition, the firing rates between dorsal and ventral STN are compared for all groups.

		No MNS	MNS-5	MNS-20	MNS-90
	Mean ± SD	54.9 ± 24.4	57.6 ± 24.2	57.8 ± 26.8	56.8 ± 24.3
Dorsal	Different percentage (%) ^1^		4.8	5.1	3.3
	*p*-Value		0.350	0.348	0.395
Ventral	Mean ± SD	60 ± 30.3	58.4 ± 35.3	57.7 ± 34.1	52.8 ± 36.0
	Different percentage (%) ^1^		−2.7	−3.8	−3.8
	*p*-Value		0.433	0.406	0.228
*p*-Value		0.261	0.463	0.498	0.322

Data are expressed as mean ± standard deviation. Different percentage (%) ^1^: ((Mean _MNS-5, MNS-20 or MNS-90_ − Mean _No MNS_)/Mean _No MNS_) × 100%.

**Table 4 biomedicines-10-00074-t004:** Spearman’s correlation between power bands, with beta and gamma and motor clinical outcomes from UPDRS.

	Part II				Part III		
	Frequency of Stimulation	Beta	Gamma	Firing Rate	Beta	Gamma	Firing Rate
		ρ/*P*	ρ/*P*	ρ/*P*	ρ/*P*	ρ/*P*	ρ/*P*
Dorsal	No MNS	N.S./N.C.	N.S./N.C.	N.S./N.C.	N.S./N.C.	N.S./N.C.	N.S./N.C.
	MNS-05	**0.023/0.706**	N.S./N.C.	N.S./N.C.	**0.012/0.755**	N.S./N.C.	N.S./N.C.
	MNS-20	**0.003/0.833**	**0.028/−0.688**	N.S./N.C.	**0.032/0.675**	N.S./N.C.	N.S./N.C.
	MNS-90	**0.005/0.802**	N.S./N.C.	N.S./N.C.	N.S./N.C.	N.S./N.C.	N.S./N.C.
Ventral	No MNS	N.S./N.C.	N.S./N.C.	N.S./N.C.	N.S./N.C.	N.S./N.C.	N.S./N.C.
	MNS-05	N.S./N.C.	**0.026/−0.693**	N.S./N.C.	N.S./N.C.	N.S./N.C.	N.S./N.C.
	MNS-20	N.S./N.C.	N.S./N.C.	N.S./N.C.	N.S./N.C.	N.S./N.C.	N.S./N.C.
	MNS-90	N.S./N.C.	N.S./N.C.	N.S./N.C.	N.S./N.C.	N.S./N.C.	N.S./N.C.

Data are expressed as *p*-value/correlation coefficient; N.S./N.C. indicates no significance/no correlation, while bold values indicate significance and correlation.

## Data Availability

The data presented in this study are available on request from the corresponding author. The data are not publicly available due to privacy.

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
