# Peer review of "Median Nerve Stimulation Facilitates the Identification of Somatotopy of the Subthalamic Nucleus in Parkinson’s Disease Patients under Inhalational Anesthesia"

_biomedicines, 2021, doi:10.3390/biomedicines10010074_

Round 1

Reviewer 1 Report

This paper is well written and discusses a key point for DBS surgery, i.e. the possibility to better localize the subthalamus by using a combined approach based on micro-electrodes recordings (MERs) during peripheral stimulation. Another interesting point addressed by this paper is the critical role played by anesthesia on MERs.

I have only minor concerns:

  • the Authors should clarify the coordinates of DBS electrodes;
  • Results may be discussed at the light of some recent papers, investigating changes in oscillatory components and related measures (PAC) between different couples of electrodes, either ventral or caudal. (see, for instance, Averna et al., Clin Neurophysiol 2021, in press).

Author Response

Reviewer 1

This paper is well written and discusses a key point for DBS surgery, i.e. the possibility to better localize the subthalamus by using a combined approach based on micro-electrodes recordings (MERs) during peripheral stimulation. Another interesting point addressed by this paper is the critical role played by anesthesia on MERs.

I have only minor concerns:

Comment 1: the Authors should clarify the coordinates of DBS electrodes;

Response: We thank the reviewer for highlighting this comment. As we mentioned in our Methods, we performed post-operative CT to confirm the position of DBS electrodes. The results also showed all 11 patients have significant improvement after DBS surgery without side effects.

Comment 2: Results may be discussed at the light of some recent papers, investigating changes in oscillatory components and related measures (PAC) between different couples of electrodes, either ventral or caudal. (see, for instance, Averna et al., Clin Neurophysiol 2021, in press).

Response: We thank the reviewer and we already cite this reference in our discussion (page 6, line 37).

Reviewer 2 Report

A nice paper. Well written, clear, and comprehensive. 

Author Response

We would like to thank the reviewers for their extensive assessment of our manuscript, and for important and helpful comments and suggestions. 

Reviewer 3 Report

The paper is of average quality – mainly because the authors are trying to find something of statistical significance within very homogeneous data, making conclusions weak and not fully supported.

In general, the differences from any location and from any stimulation frequency were less than 7% - this is completely within range of inter-personal variability and has no clinical relevance.

In abstract – there was no MSN “during DBS” – please correct

In abstract – the attempt “to delineate the mechanism of inhalational anesthesia” by testing STN MER response to MNS does not make any sense. Did the authors try to facilitate DBS surgery or to understand anesthesia mechanisms?

In Introduction – loss of dopaminergic neurons in PD is not a “dopamine denervation” – please correct.

The numbers in the table 2 and in the text do not make sense – how does (21.4-8.5)/21.4 make 55% or (21.0-9.7)/21.0 make 61.9%? please recalculate.

Same with tremor improvement – reduction from 2.9 to 0.3 is close to 90% and from 2.2 to 0.1 – to 95%. Please recalculate.

Author Response

Reviewer 3:

The paper is of average quality – mainly because the authors are trying to find something of statistical significance within very homogeneous data, making conclusions weak and not fully supported.

Point 1: In general, the differences from any location and from any stimulation frequency were less than 7% - this is completely within range of inter-personal variability and has no clinical relevance.

Response: We really appreciate your important point. We acknowledge that this is a preliminary study with a small cohort and without another control group. Our aim is to explore how different frequencies of MNS (median nerve stimulation) influence oscillation of STN during DBS surgery. We identify a significant difference between stimulation parameters by analyzing oscillation over beta and gamma bands. We also highlight the limitation in our Discussion section.

Point 2: In abstract – there was no MSN “during DBS” – please correct

Response: We sincerely thank reviewer for reminding us this issue. We revise the sentence to " Microelectrode recording (MER) during DBS was collected either with median nerve stimulation (MNS) was delivered at 5, 20, and 90 Hz frequencies and without DBS" (page 1, line 17-19).

Point 3: In abstract – the attempt “to delineate the mechanism of inhalational anesthesia” by testing STN MER response to MNS does not make any sense. Did the authors try to facilitate DBS surgery or to understand anesthesia mechanisms?

Response: Thank you for your comment. We agree that using MNS was aimed to facilitate DBS surgery. We revise the sentence to "This paradigm may help identify an alternative method to facilitate STN identification and DBS surgery under inhalational anesthesia" (page 1, line 28-29).

Point 4: In Introduction – loss of dopaminergic neurons in PD is not a “dopamine denervation” – please correct.

Response: We really appreciate that you remind this important point. We have already revised the sentence to "Parkinson ’s disease (PD) is a neurodegenerative disease and characterized by progressively loss of dopaminergic neurons" (page 1, line 34-35).

Point 5: The numbers in the table 2 and in the text do not make sense – how does (21.4-8.5)/21.4 make 55% or (21.0-9.7)/21.0 make 61.9%? please recalculate.

Response: We feel sorry that our table is not clear. The percentage of improvement(%) in table 2 was the mean of all patients and improvement(%) was not calculated from Med off and Med on in this table. We revised the legend as " Improvement (%)1 was the mean of the percentage of improvement that calculated from Med off scores –Med on scores / Med off scores × 100%. Improvement (%)2 was the mean of the percentage of improvement that calculated from Med off and DBS off scores – Med on DBS on scores / Med off and DBS off scores × 100%." in table 2 (page3, line 39-42).

Point 6: Same with tremor improvement – reduction from 2.9 to 0.3 is close to 90% and from 2.2 to 0.1 – to 95%. Please recalculate.

Response: We feel sorry that our table is not clear. But in Table 2 the percentage of improvement in tremor score was the mean of the percentage of improvement in all patients. We revised the legend as " Improvement (%)1 was the mean of the percentage of improvement that calculated from Med off scores –Med on scores / Med off scores × 100%. Improvement (%)2 was the mean of the percentage of improvement that calculated from Med off and DBS off scores – Med on DBS on scores / Med off and DBS off scores × 100%." in table 2 (page3, line 39-42).

Round 2

Reviewer 3 Report

Unfortunately, the authors failed to respond to the issue raised in the original review and the numbers still do not make sense. In the table all the values that I mentioned earlier have not been corrected or re-calculated:

21.4-8.5/21.4x100=60.2, not 55.7

21.0-9.7/21.0x100=53.8, not 61.9

2.9-0.3/2.9x100=89.6, not 68.3

2.2-0.1/2.2x100=95.5, not 76.7

The new sentence in the abstract "Microelectrode recording (MER) during DBS was collected either with median nerve stimulation (MNS) was delivered at 5, 20, and 90 Hz frequencies and without DBS" does not make sense - please re-word it.

Same with the introduction - what is "progressively loss"?

Author Response

Point 1: Unfortunately, the authors failed to respond to the issue raised in the original review and the numbers still do not make sense. In the table all the values that I mentioned earlier have not been corrected or re-calculated:

21.4-8.5/21.4x100=60.2, not 55.7

21.0-9.7/21.0x100=53.8, not 61.9

2.9-0.3/2.9x100=89.6, not 68.3

2.2-0.1/2.2x100=95.5, not 76.7

Response: We feel sorry that our table is not clear. We revised the table legend as " The extent of improvement (%)1 was first calculated from individual patient’s Med off scores –Med on scores / Med off scores × 100% and the value was the mean and standard deviation calculated from each patient’s improvement percentage in scores. Improvement (%)2 was the mean of the percentage of DBS improvement that calculated from Med off and DBS off scores – Med on DBS on scores / Med off and DBS off scores × 100%." in table 2 (page3, line 39-42).

Point 2: The new sentence in the abstract "Microelectrode recording (MER) during DBS was collected either with median nerve stimulation (MNS) was delivered at 5, 20, and 90 Hz frequencies and without DBS" does not make sense - please re-word it.

Response: We thank reviewer for this suggestion and revised the sentence as " Microelectrode recording (MER) during DBS was collected under median nerve stimulation (MNS) was delivered at 5, 20, 90 Hz frequencies and without MNS." (page1, line 21-23).

Point 3: Same with the introduction - what is "progressively loss"?

Response: We thank reviewer for this important suggestion and have revised the sentence as " Parkinson’s disease (PD) is a neurodegenerative disease and characterized by loss of dopaminergic neurons, which causes abnormalities in the downstream basal ganglia, such as abnormal firing patterns (e.g., bursting activities) and irregular oscillatory activity. " (page 1, line 37-39).